# Uncertainty-Aware Natural Language Inference with Stochastic Weight Averaging

**Aarne Talman**[*]   **Hande Celikkanat**[*]   **Sami Virpioja**[*]
**Markus Heinonen**[†]   **Jörg Tiedemann**[*]

[*] Department of Digital Humanities, University of Helsinki
`name.surname@helsinki.fi`
[†] Department of Computer Science, Aalto University
`markus.o.heinonen@aalto.fi`

## Abstract

This paper introduces Bayesian uncertainty modeling using Stochastic Weight Averaging-Gaussian (SWAG) in Natural Language Understanding (NLU) tasks. We apply the approach to standard tasks in natural language inference (NLI) and demonstrate the effectiveness of the method in terms of prediction accuracy and correlation with human annotation disagreements. We argue that the uncertainty representations in SWAG better reflect subjective interpretation and the natural variation that is also present in human language understanding. The results reveal the importance of uncertainty modeling, an often neglected aspect of neural language modeling, in NLU tasks.

## 1 Introduction

Arguably, human language understanding is not objective nor deterministic. The same utterance or text can be interpreted in different ways by different people depending on their language standards, background knowledge and world views, the linguistic context, as well as the situation in which the utterance or text appears. This uncertainty about potential readings is typically not modeled in Natural Language Understanding (NLU) research and is often ignored in NLU benchmarks and datasets. Instead, they usually assign a single interpretation as a gold standard to be predicted by an artificial system ignoring the inherent ambiguity of language and potential disagreements that humans arrive at.

Some datasets like SNLI (Bowman et al., 2015) and MNLI (Williams et al., 2018) do, however, contain information about different readings in the form of annotation disagreement. These datasets include the labels from five different rounds of annotation which show in some cases clear disagreement about the correct label for the sentence pair. Those labeling discrepancies can certainly be a result of annotation mistakes but more commonly they arise from differences in understanding the task, the given information and how it relates to world knowledge and personal experience.

Moving towards uncertainty-aware neural language models, we present our initial results using Stochastic Weight Averaging (SWA) (Izmailov et al., 2018) and SWA-Gaussian (SWAG) (Maddox et al., 2019) on the task of Natural Language Inference. SWAG provides a scalable approach to calibrate neural networks and to model uncertainty presentations and is straightforward to apply with standard neural architectures. Our study addresses the two main questions:

- How does uncertainty modeling using SWAG influence prediction performance and generalization in NLI tasks?

- How well does the calibrated model reflect human disagreement and annotation variance?

In this paper, we first test the performance of SWA and SWAG in SNLI and MNLI tasks. We then study if adding weight averaging improves the generalization power of NLI models as tested through cross-dataset experiments. Finally, we analyse the probability distributions from SWA and SWAG to test how well the model uncertainty corresponds to annotator disagreements.

## 2 Background and Related Work

### 2.1 Uncertainty in human annotations

In a recent position paper Plank (2022) argue that instead of taking human label variation as a problem, we should embrace it as an opportunity and

take it into consideration in all the steps of the ML pipeline: data, modeling and evaluation. The paper provides a comprehensive survey of research on (i) reasons for human label variation, (ii) modeling human label variation, and (iii) evaluating with human label variation.

Pavlick and Kwiatkowski (2019) studied human disagreements in NLI tasks and argue that we should move to an evaluation objective that more closely corresponds to the natural interpretation variance that exists in data. Such a move would require that NLU models be properly calibrated to reflect the distribution we can expect and, hence, move to a more natural inference engine.

Chen et al. (2020) propose Uncertain NLI (UNLI), a task that moves away from categorical labels into probabilistic values. They use a scalar regression model and show that the model predictions correlate with human judgement.

## 2.2 Representing Model Uncertainty

The approach to uncertainty modeling that we consider is related to the well-established technique of model ensembling. Stochastic optimization procedures applied in training deep neural networks are non-deterministic and depend on hyper-parameters and initial seeds. Ensembles have been used as a pragmatic solution to average over several solutions, and the positive impact on model performance pushed ensembling into the standard toolbox of deep learning. Related to ensembling is the technique of checkpoint averaging (refer to e.g. Gao et al., 2022), which is also known to improve performance.

Intuitively, ensembles and checkpoint averages also reflect the idea of different views and interpretations of the data and, therefore, provide a framework for uncertainty modeling. Stochastic Weight Averaging (SWA, Izmailov et al. (2018)) and SWA-Gaussian (SWAG, Maddox et al. (2019)) both build on this idea. SWA proposes using the first moments of the parameters of the solutions traversed by the optimizer during the optimization process, as mean estimates of the model parameters. Using such mean values have been argued to result in finding wider optima, providing better generalization to unseen data. On top of these mean estimations procured by SWA, SWAG then adds a low-rank plus diagonal approximation of covariances, which, when combined with the aforementioned mean estimations, provide us

with corresponding Gaussian posterior approximations over model parameters. Posterior distribution approximations learned this way then represent our epistemic uncertainty about the model (Kiureghian and Ditlevsen, 2009), meaning the uncertainty stemming from not knowing the perfect values of the model parameters, since we do not have infinite data to train on. During test time, instead of making estimates from a single model with deterministic parameters, we sample $N$ different models from the approximated posteriors for each model parameter, and use the average of their prediction distributions as the model response.

Note that as both of these methods use the optimizer trajectory for the respective approximations, they provide significant computational efficiency as compared to the vanilla ensembling baseline. In this paper, we use SWA mainly as another baseline for SWAG, which needs to outperform SWA in order to justify the additional computation required for the covariance approximation.

SWA (Izmailov et al., 2018) is a checkpoint averaging method that tracks the optimization trajectory for a model during training, using the average of encountered values as the eventual parameters:

$$\theta_{\text{SWA}} = \frac{1}{T} \sum_{i=1}^{T} \theta_i, \qquad (1)$$

with $\theta_{\text{SWA}}$ denoting the SWA solution for parameter $\theta$ after T steps of training.[1]

SWAG (Maddox et al., 2019) extends this method to estimate Gaussian posteriors for model parameters, by also estimating a covariance matrix for the parameters, using a low-rank plus diagonal posterior approximation. The diagonal part is obtained by keeping a running average of the second uncentered moment of each parameter, and then at the end of the training calculating:

$$\Sigma_{\text{diag}} = \text{diag}(\frac{1}{T} \sum_{i=1}^{T} \theta_i^2 - \theta_{\text{SWA}}^2) \qquad (2)$$

while the diagonal part is approximated by keeping a matrix $DD^{\top}$ with columns $D_i = (\theta_i - \hat{\theta}_i)$, $\hat{\theta}_i$ standing for the running estimate of the parameters' mean obtained from the first $i$ samples. The rank of the approximation is restricted by keeping only the final K-many of the $D_i$ vectors, and dropping the previous, with K being a hyperparameter

---

[1]In this work, we use one sample from each epoch.

of the method:

$$\Sigma_{\text{low-rank}} \approx \frac{1}{K-1}DD^{\mathsf{T}} \tag{3}$$

$$= \frac{1}{K-1} \sum_{i=T-K+1}^{T} (\theta_i - \hat{\theta}_i)(\theta_i - \hat{\theta}_i)^{\mathsf{T}} \tag{4}$$

The overall posterior approximation is given by:

$$\theta_{\text{SWAG}} \sim \mathcal{N}(\theta_{\text{SWA}}, \frac{1}{2}(\Sigma_{\text{diag}} + \Sigma_{\text{low-rank}})). \tag{5}$$

Once the posteriors are thus approximated, in test time, the model is utilized by sampling from the approximated posteriors for $N$ times, and taking the average of the predicted distributions from these samples as the answer of the model.

One of the advantages of SWAG is the possibility to seamlessly start with any pre-trained solution. Approximating the posterior is then done during fine-tuning without the need to change the underlying model.

## 2.3 Stochastic Weight Averaging in NLP

Previous work on Stochastic Weight Averaging in the context of NLP is very limited. Lu et al. (2022) adapt SWA for pre-trained language models and show that it works on par with state-of-the-art knowledge distillation methods. Khurana et al. (2021) study pre-trained language model robustness on a sentiment analysis task using SWA and conclude that SWA provides improved robustness to small changes in the training pipeline. Kaddour et al. (2022) test SWA on multiple GLUE benchmark tasks (Wang et al., 2018) and find that the method does not provide clear improvement over the baseline.

Maddox et al. (2019) test SWAG in language modeling tasks using Penn Treebank and WikiText-2 datasets and show that SWAG improves test perplexities over a SWA baseline. To the best of our knowledge our work is the first to apply SWAG to NLU tasks.

## 3 Experiments

We test the performance of SWA and SWAG on the natural language inference task using three NLI datasets, including cross-dataset experiments, and study the effect on both hard and soft labeling. Code for replicating the experiments is available on GitHub: `https://github.com/Helsinki-NLP/uncertainty-aware-nli`

## 3.1 Datasets

We use Stanford Natural Language Inference corpus (SNLI) (Bowman et al., 2015) and Multi-Genre Natural Language Inference (MNLI) corpus (Williams et al., 2018) as the datasets in our experiments. We also study cross-dataset generalisation capability of the model with and without weight averaging. For those experiments we also include SICK (Marelli et al., 2014) as a test set. In cross-dataset generalization experiments we first fine-tune the model with a training data from one NLI dataset (e.g. SNLI) and then test with a test set from another NLI dataset (e.g. MNLI-mm).

**SNLI** The Stanford Natural Language Inference (SNLI) corpus is a dataset of 570k sentence pairs which have been manually labeled with entailment, contradiction, and neutral labels. The source for the premise sentences in SNLI were image captions from the Flickr30k corpus (Young et al., 2014).

**MNLI** The Multi-Genre Natural Language Inference (MNLI) corpus is made of 433k sentence pairs labeled with entailment, contradiction and neutral, containing examples from ten genres of written and spoken English. Five of the genres are included in the training set. The development and test sets have been split into matched (MNLI-m) and mismatched (MNLI-mm) sets, where the former includes only sentences from the same genres as the training data, and the latter includes genres not present in the training data.[2] The MNLI dataset was annotated using very similar instructions as for the SNLI dataset and, therefore it is safe to assume that the definitions of entailment, contradiction and neutral are the same across these two datasets.

**SICK** SICK is a dataset that was originally designed to test compositional distributional semantics models. The dataset includes 9,840 examples with logical inference (negation, conjunction, disjunction, apposition, relative clauses, etc.). The

---

[2]As the test data for MNLI have not been made publicly available, we use the development sets when reporting the results for MNLI.

dataset was constructed automatically by taking pairs of sentences from a random subset of the 8K ImageFlickr (Young et al., 2014) and the SemEval 2012 STS MSRVideo Description (Agirre et al., 2012) datasets by using rule-based approach to construct examples for the different logical inference types.

## 3.2 Methods

In all the experiments we fine tune a pre-trained RoBERTa-base model (Liu et al., 2019) from the Hugging Face Transformers library (Wolf et al., 2020). As a common practice in the NLI tasks, we use the majority-vote gold labels for training.

We add stochastic weight averaging to the RoBERTa model by using the SWA implementation from PyTorch 1.12[3] and the SWAG implementation by (Maddox et al., 2019)[4]. To study how well SWA and SWAG perform in NLI as compared to a baseline model, we ran the same fine-tuning with SNLI and MNLI datasets, while utilizing SWA and SWAG for mean and variance estimations of parameters undergoing fine-tuning.

## 3.3 Results

The standard evaluation for the NLI task is the accuracy on aggregated gold labels. However, as two of the test data sets (from SNLI and MNLI) also contains multiple human annotations, we also use those for measuring the cross entropy of the predicted distribution on the human label distribution (soft labeling, e.g. Peterson et al., 2019; Pavlick and Kwiatkowski, 2019).

### 3.3.1 Accuracy

The basic classification results are in Table 1. We report average accuracies and standard deviation over 5 runs with different random seeds.

Both SWA and SWAG provide clear improvements over the baseline without weight averaging. SWAG performs slightly better than SWA across all the three experiments.

In order to test if weight averaging improves the generalization capability of NLI models, we further performed cross-dataset generalization tests

---

[3] https://pytorch.org/docs/1.12/optim.html#stochastic-weight-averaging

[4] https://github.com/wjmaddox/swa_gaussian

| Dataset | Method | Acc (%) | SD | Δ |
|---|---|---|---|---|
| SNLI | base | 90.80 | 0.26 | - |
| SNLI | SWA | 91.47 | 0.24 | +0.67 |
| SNLI | SWAG | **91.59** | 0.14 | **+0.79** |
| MNLI-m | base | 86.53 | 0.20 | - |
| MNLI-m | SWA | 87.60 | 0.19 | +1.07 |
| MNLI-m | SWAG | **87.76** | 0.12 | **+1.23** |
| MNLI-mm | base | 86.31 | 0.26 | - |
| MNLI-mm | SWA | 87.34 | 0.29 | +1.03 |
| MNLI-mm | SWAG | **87.51** | 0.19 | **+1.20** |

Table 1: Comparison of SWA and SWAG performance on NLI benchmarks (mean accuracy and standard deviation over 5 runs). Δ is the difference to the baseline result (base) with no weight averaging.

| Dataset | Method | Acc (%) | SD | Δ |
|---|---|---|---|---|
| SNLI → MNLI-m | base | 77.31 | 0.57 | |
| SNLI → MNLI-m | SWA | **79.67** | 0.37 | **2.36** |
| SNLI → MNLI-m | SWAG | 79.33 | 0.21 | 2.02 |
| SNLI → MNLI-mm | base | 77.40 | 0.78 | |
| SNLI → MNLI-mm | SWA | **79.44** | 0.19 | **2.04** |
| SNLI → MNLI-mm | SWAG | 79.24 | 0.29 | 1.84 |
| SNLI → SICK | base | 57.08 | 0.77 | |
| SNLI → SICK | SWA | 57.09 | 0.32 | 0.01 |
| SNLI → SICK | SWAG | **57.17** | 0.37 | **0.08** |
| MNLI → SNLI | base | 82.84 | 0.74 | |
| MNLI → SNLI | SWA | 84.15 | 0.35 | 1.31 |
| MNLI → SNLI | SWAG | **84.45** | 0.27 | **1.61** |
| MNLI → SICK | base | **56.63** | 0.94 | |
| MNLI → SICK | SWA | 56.17 | 0.60 | -0.46 |
| MNLI → SICK | SWAG | 56.53 | 0.91 | -0.10 |

Table 2: Cross-dataset experiments with and without weight averaging (mean accuracy and standard deviation over 5 runs with different random seeds), where the left hand side of the arrow is the training set and the right hand side is the testing set.

following (Talman and Chatzikyriakidis, 2019). The results are reported in Table 2.

The results of cross-dataset experiments are slightly mixed: We do not notice a clear advantage of SWAG over SWA, but with the exception of training with MNLI and testing with SICK, we do notice improvement for weight averaging approaches as compared to the baseline. The performance on SICK drops significantly in all cases and the difference between the approaches is minimal, showing that the NLI training data is not a good fit for that benchmark. The other cross-dataset results highlight the advantage of stochastic weight averaging, which is in line with the findings of (Izmailov et al., 2018) that the method is able to locate wider optima regions with better generalization capabilities.

| Dataset | Method | Cross Entropy | Δ |
|---|---|---|---|
| SNLI | base | 0.83 | |
| SNLI | SWA | 0.75 | -0.08 |
| SNLI | SWAG | **0.69** | **-0.14** |
| MNLI-m | base | 0.87 | |
| MNLI-m | SWA | 0.80 | -0.07 |
| MNLI-m | SWAG | **0.73** | **-0.14** |
| MNLI-mm | base | 0.84 | |
| MNLI-mm | SWA | 0.77 | -0.07 |
| MNLI-mm | SWAG | **0.69** | **-0.15** |
| SNLI → MNLI-m | base | 1.13 | |
| SNLI → MNLI-m | SWA | 0.90 | -0.23 |
| SNLI → MNLI-m | SWAG | **0.80** | **-0.33** |
| SNLI → MNLI-mm | base | 1.12 | |
| SNLI → MNLI-mm | SWA | 0.88 | -0.24 |
| SNLI → MNLI-mm | SWAG | **0.79** | **-0.33** |
| MNLI → SNLI | base | 1.04 | |
| MNLI → SNLI | SWA | 0.97 | -0.07 |
| MNLI → SNLI | SWAG | **0.89** | **-0.15** |

Table 3: Comparison of cross entropies between data annotation distributions using base, SWA and SWAG methods. Δ is the difference to the baseline cross entropy values.

### 3.3.2 Cross Entropy

We also test how well these weight averaging and covariance estimating methods help towards better modeling annotator disagreement and annotation uncertainty in the NLI testsets of SNLI and MNLI. These two datasets come with five different annotation labels for every data point, often with high disagreement between human annotators, indicating inherently *confusing* data points with high aleatoric uncertainty (Kiureghian and Ditlevsen, 2009). For quantifying the goodness of fit of the model predictions, we calculate the cross entropy between the predicted and annotation distributions.[5]

Table 3 depicts the resulting cross entropy values, with lower values denoting more faithful predictions. SWA and SWAG result in consistently more similar distributions with that of annotations, complementing their overall better accuracy results (Section 3.3). In contrast to the accuracy results, here SWAG outperforms SWA in all cases, indicating that the modeling uncertainty through the approximation of Gaussian posteriors helps to model annotator disagreements more accurately. The results also carry over to the cross-dataset experiments as shown on the table.

The comparison between system predictions

and annotator variation deserves some further analysis. Preliminary study (refer to examples in Appendix A) indicates that the prediction uncertainty in SWAG for individual instances very well follows human annotation confusion. Furthermore, we identified cases with a larger mismatch between system predictions and human disagreement where the latter is mainly caused by erroneous or at least questionable decisions. This points to the use of SWAG in an active learning scenario, where annotation noise can be identified using a well calibrated prediction model.

## 4 Conclusions

Our results show that weight averaging provides consistent and clear improvement for both SNLI and MNLI datasets. The cross-dataset results are slightly mixed but also show the trend of improved cross-domain generalization. Finally, we demonstrate a clear increase in the correlation with human annotation variance when comparing SWAG with non-Bayesian approaches.

For future work we consider making use of multiple annotations also during training and extensions of SWAG such as MultiSWAG (Wilson and Izmailov, 2020). We also plan to test the methods on different NLU datasets, especially those with a high number of annotations (e.g. Nie et al., 2020), and compare the annotation variation and system predictions in more detail. Finally, in our future work we will explore other uncertainty modeling techniques, like MC dropout (Gal and Ghahramani, 2016), in NLU and see how they compare with stochastic weight averaging techniques.

## Acknowledgements

This work is supported by the ICT 2023 project "Uncertainty-aware neural language models" funded by the Academy of Finland (grant agreement 345999) and the FoTran project, funded by the European Research Council (ERC) under the European Unions Horizon 2020 research and innovation programme (grant agreement no. 771113). We also wish to acknowledge CSC – The Finnish IT Center for Science for the generous computing resources they have provided.

---

[5]Note that for the Baseline and SWA models, we consider the output from the eventual softmax function as the predicted distribution, while for the SWAG model, we use the average output distribution from $N = 20$ sampled models.

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

# A    Appendix

Here we showcase and discuss three randomly selected data points from the SNLI dataset, and compare the predictions of the $N = 20$ samples from the SWAG model with the annotation distributions for each of these points. Table 4 presents two cases (left and middle) in which the SWAG model makes the correct prediction, and another case (right) in which the model makes an incorrect prediction. In the high agreement cases, indicated by lower cross entropies between the annotations and prediction, the SWAG model not only selects the correct label for the instance, but also predicts the annotator disagreement correctly when such a disagreement exists (middle) versus when it does not (left).

The third figure presents a case where the predictions of the SWAG samples are more *certain* than expected: Annotators disagree on whether the hypothesis is entailment or neutral, whereas the model predictions place all probability mass to the neutral class. The corresponding cross entropy is high, which reflects this disagreement. It should be noted that this is also a fairly controversial and difficult data point, and to conclude Entailment requires making some strong assumptions. Ideally, such disagreements between system predictions and annotator distributions may also be used as cues within the training process itself. Two potential venues are (1) using the incongruence between the two distributions as the loss signal to drive the optimization process directly (as opposed to using only the gold label and the predicted class label), and (2) using the incongruence in predictions in an active learning scenario.

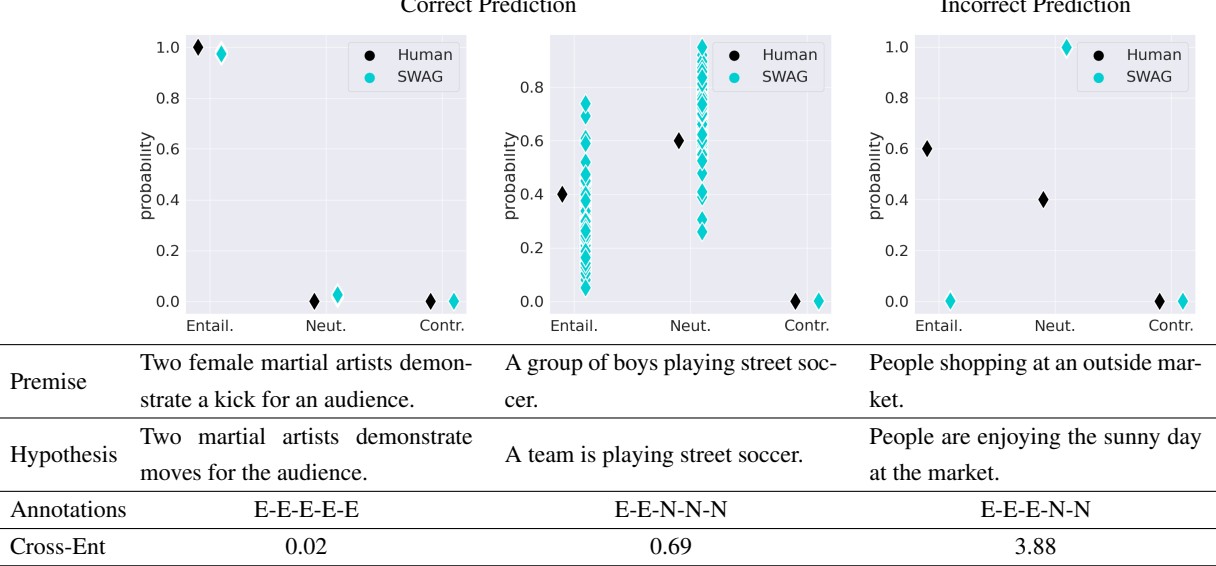

| | Correct Prediction | | Incorrect Prediction |
|---|---|---|---|
| Premise | Two female martial artists demonstrate a kick for an audience. | A group of boys playing street soccer. | People shopping at an outside market. |
| Hypothesis | Two martial artists demonstrate moves for the audience. | A team is playing street soccer. | People are enjoying the sunny day at the market. |
| Annotations | E-E-E-E-E | E-E-N-N-N | E-E-E-N-N |
| Cross-Ent | 0.02 | 0.69 | 3.88 |

Table 4: Comparison of probability distributions of human annotations vs. SWAG model predictions, for three randomly selected data points from the SNLI dataset. *(Left and middle)* Correctly predicted cases, as indicated by low cross entropy, *(Right)* A incorrectly predicted case, as indicated by high cross entropy. SWAG points indicate the outputted probability distributions from $N = 20$ samples.