# OpenReview forum: "Uncertainty-Aware Natural Language Inference with Stochastic Weight Averaging"
_NoDaLiDa/2023/Conference — NoDaLiDa 2023_

### Official Review · Reviewer_YuTc · 2023-03-09
**Review: Uncertainty-Aware Natural Language Inference with Stochastic Weight Averaging**

**Rating:** 7
**Confidence:** 5

**Review:**

This paper trains, and evaluates transformer-based NLI models (using SNLI, MNLI, and SICK) that have been trained to model uncertainty using SWA and SWAG. The evaluation are performed on cross-datasets (i.e. trained on one dataset and tested on another). To evaluate these models they are comparing the cross-entropy between the predicted labels and the variance of labels the annotators provided. The methods the authors are using appears to better model the annotator uncertainty than the baselines.

The paper is interesting and continues a line of questioning often neglected in NLU. However, there are some things that stand out in particular.

* Why is SICK included in the tests of table 2? I thought the main research question was to investigate uncertainty, and the results for SICK does not address that research question. Rather, it's only helps showcase the general effectiveness of the method, which is fine, but a bit confusing given the main purpose of the paper.
* The paper presents a very interesting research question, but the analysis appears to be lacking. It would be interesting to see a bit in more detail how the model performs given the uncertainty of the annotators. For example, are there any patterns of the annotations which the model struggle for? i.e. is it more difficult for models to predict annotator uncertainty when the majority class is Entailment? or Contradiction?
* Why do you think the SWA and SWAG helps predict uncertainty? I guess for a short paper an extensive analysis is not reasonable, but I was expecting at least 2-3 sentences speculating about this.

I get that this is a short paper, but it's also very interesting, so I'm a bit annoyed that there is not a more detailed analysis. My suggestion would be to skip the SICK dataset, maybe describe SWA/SWAG more briefly and add some more analysis given the patterns of annotator uncertainty.


**Paper Type:**

Short paper

---

### Official Review · Reviewer_ZRVo · 2023-03-12
**A solid set of experiments on SWA with some points of improvement for camera ready**

**Rating:** 7
**Confidence:** 5

**Review:**

This paper applies Stochastic Weight Averaging and Stochastic Weight Averaging-Gaussian to two NLI datasets: MNLI and SNLI. The paper shows that an increase in predictive performance with both techniques and that SWA and SWAG can be advantageous for cross-dataset accuracy in some cases. Additionally, the paper shows that SWAG is closer to modeling the distribution of annotations than base or SWA models, benefitting from its uncertainty-aware nature.

The paper has the following strengths:

The paper provides a clear motivation for why SWAG’s impact on the task of NLI is studied. It highlights important limitations of most NLP research: most tasks model a specific type of belief and leave behind the ambiguity of language. The setup of experiments is very clear, it is also very nice that the paper clearly mentions the standard deviation and how many seeds were used. The paper is well-written and has a clear structure.

The following points would improve the paper:

- Using SWA for uncertainty is not well motivated, the relation is unclear. It would be nice if you could link the nature of SWA to annotator (dis)agreement. A pretty popular and easy-to-implement way of measuring uncertainty in large language models is using MC Dropout. It would be nice to add the results with this technique as a comparison/baseline. At least, a small explanation of how MC Dropout and SWAG are different from each other in the related work would be a valuable addition. If this is too complicated or takes too much space, maybe leave the reference to disagreement out of the paper. The experiments on SWA(G) can stand on their own for a short paper.
- I also recommend make it clear that taking checkpoints for SWA after every epoch is a design choice and that this could also be done after any chosen timestep (e.g. every 200 steps) according to the original algorithm.
- The Related Work would benefit from a section on work done in NLP w.r.t. SWA(G). There are not many publications on this and it would be good if those that are there are known to researchers working with SWA(G) in NLP. An overview of work known to me can be found below.
- The Related Work Section on SWA(G) could use an explanation of what sets this technique apart from other weight-averaging techniques.
- While it does seem that SWA and SWAG improve the predictive performance, calling it “significant”, especially in the case of SWA, might be too optimistic when the increase in accuracy is around 1% and there are no significance tests done.
- The last sentence of 3.3.1 says that weight averaging is advantageous as shown by the cross-dataset results and that improved modeling of uncertainty can lead to better generalizations. Does this specifically applies to SWAG but not SWA since the latter does not have a direct mechanism of uncertainty? It would be good to clarify this.
- The current explanation of SWA(G) in the Experiments section has low reproducibility. The paper would benefit from a Technical Details section (this can be placed in the Appendix) on how SWA and SWAG are employed, when do you start averaging the checkpoints, i.e. when do you switch to the SWA(G) learning schedule?

Verification question: Is the SWA model (\theta_SWA) in Equation 3 the SWA model of Epoch T or Epoch T-1?
Any ideas on why there is no clear advantage of SWAG over SWA for the cross-dataset experiments?

Minor points:

Table 2:
SNLI -> SICK for SWAG $\delta$ should be 0.9 instead of 0.8
SNLI -> MNLI-m SWA $\delta$ should be 2.36 instead of 2.37
SNLI -> MNLI-m SWAG $\delta$ should be 2.02 instead of 2.03

Related work on SWA in NLP:

Guo, H., Jin, J. and Liu, B., 2023. Stochastic weight averaging revisited. Applied Sciences, 13(5), p.2935.

Kaddour, J., Liu, L., Silva, R. and Kusner, M., When Do Flat Minima Optimizers Work?. In Advances in Neural Information Processing Systems.

Khurana, U., Nalisnick, E. and Fokkens, A., 2021, November. How Emotionally Stable is ALBERT? Testing Robustness with Stochastic Weight Averaging on a Sentiment Analysis Task. In Proceedings of the 2nd Workshop on Evaluation and Comparison of NLP Systems (pp. 16-31).

Lu, P., Kobyzev, I., Rezagholizadeh, M., Rashid, A., Ghodsi, A. and Langlais, P., 2022. Improving Generalization of Pre-trained Language Models via Stochastic Weight Averaging. In Findings of EMNLP.

**Paper Type:**

Short paper

---

### Decision · Program_Chairs · 2023-03-17

Accept